# The Influence of Sex on Characteristics and Outcomes of Coronavirus-19 Patients: A Retrospective Cohort Study

**DOI:** 10.3390/jcm12031118

**Published:** 2023-01-31

**Authors:** Song-I Lee, Chaeuk Chung, Dongil Park, Da Hyun Kang, Ye-Rin Ju, Jeong Eun Lee

**Affiliations:** Division of Pulmonary and Critical Care Medicine, Department of Internal Medicine, Chungnam National University Hospital, Chungnam National University School of Medicine, Daejeon 35015, Republic of Korea

**Keywords:** COVID-19, sex, outcome

## Abstract

Background: The influence of sex on the clinical characteristics and prognosis of coronavirus disease (COVID-19) patients is variable. This study aimed to evaluate COVID-19 management based on sex differences. Methods: We retrospectively reviewed COVID-19 patients who were admitted to the tertiary hospital between January 2020 and March 2021. Logistic regression analysis was used to evaluate the factors associated with in-hospital mortality. Results: During the study period, 584 patients were admitted to our hospital. Among them, 305 patients (52.2%) were female, and 279 patients (47.8%) were male. Males were younger than females, and frailty scale was lower in males than in females. Fever was more common in males, and there was no difference in other initial symptoms. Among the underlying comorbidities, chronic obstructive disease was more common in males, and there were no significant differences in other comorbidities. Moreover, treatment, severity, and outcome did not significantly differ between the groups. The risk factors for in-hospital mortality were age, high white blood cell count, and c-reactive protein level. Conclusions: We found no definite sex differences in the clinical characteristics and outcomes of COVID-19 patients. However, a better understanding of sex-dependent differences in COVID-19 patients could help in understanding and treating patients.

## 1. Introduction

The viral infection caused by severe acute respiratory syndrome coronavirus 2 is named “coronavirus disease-19 (COVID-19)” and the World Health Organization declared a pandemic on 11 March 2020 [1]. On 3 August 2022, the Republic of Korea reported 20,052,305 diagnosed cases of COVID-19 and 25,110 deaths. In addition, by 20 July 2022, 128,050,297 vaccines were administered [2]. Most people infected with COVID-19 had some light symptoms and were able to remain at home with symptomatic treatment or antiviral medications. Hospitalized COVID-19 patients could be treated with steroids [3], remdesivir [4], IL-6 inhibitors [5], and baricitinib (janus kinase inhibitors) [6] depending on the severity [7].

The basic difference between humans is their sex, i.e., male or female. Some studies have shown that the disease severity and patient mortality of COVID-19 are higher in males than in females [8,9,10,11]. In a Danish nationwide study by Kragholm et al., the male sex showed an association with higher mortality, severity, and intensive care unit (ICU) admission than the female sex [10]. In Peckham’s meta-analysis, males showed a higher probability of needing an intensive treatment unit and a higher mortality rate than females [11]. Lee’s retrospective cohort study conducted in South Korea showed that males required more oxygen therapy and ICU admissions than females; however, there was no association with mortality [12]. In addition to sex, older age [13,14], comorbidities [15,16], abnormal laboratory findings [17,18,19], and higher viral ribonucleic acid levels [20] are known to be related with the severity and mortality in COVID-19 patients.

Sex and other clinical factors have an association with patient prognosis. However, there are few studies that have investigated the differences in symptoms, underlying conditions, treatment, and prognosis based on sex. Therefore, we investigated the clinical features and prognosis of patients according to their sex in this study.

## 2. Materials and Methods

This study retrospectively collected and analyzed the data from all patients with COVID-19 who were admitted to the tertiary care hospital in South Korea from 2 February 2020 to 31 March 2021. The ethical committee of the center (Institutional Review Board of Chungnam National University Hospital; approval no. 2021-04-053) approved this study. Because of the retrospective nature of this study, written informed consent was not needed.

Data were collected from electronic medical records (C&U Care 2.0). The patients’ basic characteristics, symptoms before admission, and laboratory findings at admission were collected. We also extracted data on received medical treatment and invasive treatment (arterial line, central line, mechanical ventilation, tracheostomy, continuous renal replacement therapy [CRRT]), the do-not-resuscitate (DNR) document, in-hospital mortality, and duration of hospital stay. The patient’s condition at admission was assessed by analyzing the Sequential Organ Failure Assessment (SOFA) score to evaluate the patient’s severity and the Clinical Fatigue Scale (CFS) score for the assessment of weakness.

### 2.1. Definition

The diagnosis of COVID-19 was confirmed by polymerase chain reaction. Severe infection was defined as SpO2 < 94% on room air, respiratory rate > 30 breaths/min, a ratio of partial pressure of oxygen to fraction of inspired oxygen (PaO2/FiO2) < 300 mmHg in arterial blood gas analysis, or lung infiltration > 50% in imaging studies. A patient showing respiratory failure, multiple organ dysfunction, or septic shock was classified as having a critical infection [21].

### 2.2. Statistical Analysis

All data and values are expressed as percentages for categorical variables or as median (interquartile range [IQR]: 25th–75th percentile) and mean ± standard deviation for continuous variables. We analyzed continuous data using Student’s t-test and categorical data using Pearson’s chi-square test or Fisher’s exact test. Logistic regression analysis was performed for evaluating the risk factors for in-hospital mortality. In univariate analysis, factors with *p* < 0.1 were identified, and multivariate analysis was performed with these factors. We used odds ratios (ORs) and 95% confidence intervals (CIs) to represent risk factors of in-hospital mortality. The *p*-value of <0.05 was considered statistically significant. We performed statistical analysis using the Statistical Package for the Social Sciences software (version 22.0; IBM Corporation, Somers, NY, USA).

## 3. Results

### 3.1. Characteristics and Clinical Features of COVID-19 Patients

In total, 584 patients were admitted to our hospital. Among them, 279 (47.8%) were male and 305 (52.2%) were female.

The characteristics and clinical features of enrolled patients are shown in Table 1. The male group was younger (55.0 (39.0–64.0) vs. 58.0 (48.0–68.0) years, *p* < 0.001), the CFS was lower (1.9 ± 1.0 vs. 2.2 ± 1.2, *p* = 0.003), and the body mass index (BMI) was higher (24.8 (22.9–27.3) vs. 23.9 (21.7–26.2) kg/m^2^, *p* < 0.001) than the female group. Regarding pre-hospitalization symptoms, the male group showed greater incidence of fever and less incidence of sore throat than the female group.

The baseline comorbidities and laboratory findings of the patients are presented in Table 2. There was no significant statistical difference in the underlying diseases, except that the incidence of chronic obstructive lung disease (COPD) was higher in the male group (3.9% vs. 1.3%, *p* = 0.045). In the laboratory findings, the male group had higher white blood cells (WBCs), total bilirubin, alanine aminotransferase (ALT), creatinine, and c-reactive protein (CRP), and, conversely, the D-dimer level was lower in the female group.

### 3.2. Treatment and Prognosis of COVID-19 Patients

The treatment and prognosis of patients are shown in Table 3. There was little difference in the application of vasopressors or the type of vasopressor (norepinephrine, vasopressin, or dobutamine) among the two groups. Moreover, there was no statistical difference between the two groups in the devices that received oxygen supply (nasal prong, high-flow nasal cannula, invasive mechanical ventilation, and extracorporeal membrane oxygenation). Medical treatment (steroids, antibiotics, and remdesivir) also showed no statistically significant difference.

In addition, there was no statistically significant difference in the presence of severe infection (6.8% vs. 9.5%, *p* = 0.236) or critical infection (10.0% vs. 8.2%, *p* = 0.440) between the two groups. Regarding the invasive treatment, there was no statistical difference in terms of whether arterial line, central line, tracheostomy, or CRRT were performed. There was no statistically significant difference in in-hospital mortality (2.9% vs. 4.3%, *p* = 0.366) between the two groups. In addition, the length of hospital stays and DNR between the two groups showed no statistical difference.

### 3.3. Factors Affecting In-Hospital Mortality in COVID-19 Patients

Figure 1 shows the Kaplan–Meier survival curves of patient groups according to sex. The male group seemed to have higher survival, but it was not statistically significant (*p* = 0.359). Table 4 shows the factors affecting in-hospital mortality using the multivariate logistic regression model. After regulating for confounders, the meaningful factors of in-hospital mortality included age (OR: 1.151, 95% CI: 1.089–1.217; *p* < 0.001), high WBC (OR: 1.489, 95% CI: 1.185–1.872; *p* = 0.001), and high CRP (OR: 1.137, 95% CI: 1.054–1.226; *p* = 0.054). Male sex was not an independent factor for in-hospital mortality in this study.

## 4. Discussion

We conducted a study to determine the effect of sex on the clinical characteristics and prognosis of COVID-19 patients. In this research, 47.8% of the hospitalized COVID-19 patients were males. Males were younger, had lower CFS, and had a higher BMI than females. COPD was the more common underlying disease in males. In other studies, the age differences between males and females vary. There were studies in which males were older than females [10,22], male and female ages did not show a statistically significant difference [12,23], and females were older than males. However, in our study, males were younger, which may have been influenced by the longer life expectancy of women in South Korea [24]. In the study of Fortunato et al., diabetes was more common in males than in females [23]. Kragholm et al. showed that underlying diseases such as sleep apnea, hypertension, diabetes, prior myocardial infarction, chronic ischemic heart disease, and chronic renal disease were more frequent in males than in females [10]. A study by Barelling et al. showed no significant difference in chronic diseases and smoking between the sexes [22]. The differences between prehospital symptoms and laboratory findings have rarely been compared in detail in previous studies. In this study, males and females were compared, and fever was the most common prehospital symptom in males; laboratory findings showed leukocytosis and hyperbilirubinemia, higher ALT, creatinine, and CRP levels, and lower D-dimer levels in males than in females. A study by Statsenko et al. showed that ALT, aspartate aminotransferase, and lactate dehydrogenase were elevated in males, and sore throat and headache were twice as common in females [25].

Male sex is thought to be a factor associated with disease severity and poor prognosis in COVID-19 patients. However, in this study, the severity of disease, invasive treatment, length of hospital stay, and in-hospital mortality did not differ between males and females. Additionally, the male sex alone was not associated with in-hospital mortality. Age and high WBC and CRP levels were associated with the prognosis. Although this study did not show a difference in mortality between males and females, other studies have noted a poor prognosis in males. In a study from the Netherlands, the odds ratio of male–female for case fatality was 1.33, and the same result was shown after adjusting for age and underlying disease [26]. In Alwani et al.’s review, although it varied by country, mortality tended to be higher in males, and ICU admission was more common in males [27]. However, Ahrenfeldt et al.’s study showed that the sex difference for death in most Europe regions increased in those under the age of 69, but then decreased, and the sex difference was the smallest in those over the age of 80 [28]. There are several factors associated with mortality in hospitalized COVID-19 patients in other studies: malnutrition, old age, immunosuppression status, underlying comorbidities, and patient severity [29,30,31]. Kahn et al. studied the severity of COVID-19 and the association between vaccination state and other baseline conditions. Vaccination was an important factor in preventing progression to severe disease and showed benefits regardless of sex. In unvaccinated patients, severity was higher in older people and men with two or more underlying diseases [32]. Differences in immune responses and lifestyle between men and women are thought to be possible; however, there is unsatisfactory information to support this hypothesis. First, it is known that the viral immune response of females is stronger than that of males, and it is conceivable that the composition of the X and Y chromosomes in women and men may be different [33]. These differences in sex hormones and X chromosomes are believed to affect the expression of transmembrane protease serine subtype 2 (TMPRSS2) and angiotensin-converting enzyme 2 (ACE2), which are associated with viral infection [34]. In addition, drinking and smoking are more common in males [35,36], and this may be related to underlying comorbidities such as COPD, heart disease, and cancer [37,38,39,40]. Smoking is also associated with high ACE2 [41]. Despite these differences, there were no differences in outcomes according sex and sex hormones in this study and other studies conducted in South Korea [12]. Therefore, the effect of these differences according to sex is unclear, and further studies are required.

Previous studies have analyzed the sex differences according to the disease, in which the differences between patients according to sex during influenza epidemics have been analyzed [42,43,44]. In a prospective multicenter study of influenza-positive patients by Karolyi et al. [43], males were younger (70 (60–79) vs. 76 (62–85) years, *p* < 0.001) and more likely to be smokers (37.9% vs. 20%, *p* < 0.001), and chronic liver disease (8.8% vs. 2.9%, *p* = 0.006) was more common than in females. In the laboratory data, creatinine, creatinine-kinase, ALT, gamma-glutamyl-transferase, and bilirubin were higher in the male group, and counts of thrombocyte were lower. In addition, although males had more ICU admissions than females, there was no significant difference in 90-day mortality within the hospital. In Jin et al.’s study of influenza patients, sex was not statistically associated with the mortality burden of influenza [44]. In addition, there have been studies on the effect of sex on acute respiratory distress syndrome (ARDS), which can appear in a severe form in COVID-19. In Heffernan et al.’s study of ARDS patients with trauma, the incidence of ARDS was higher in women; however, there was no association of sex with mortality [45]. A study of ARDS patients from 1979 to 1996 showed consistently higher mortality rates in males than in females [46]. Additionally, several studies have been conducted on the effect of sex on sepsis, which may appear to be similar to the immune response that occurs in COVID-19. Even in a systematic review of sex-associated outcomes in sepsis patients, the effect of sex-dependent mortality is not clear, and the possibility that mortality in women may be slightly higher has been mentioned [47]. In this way, the possibility of differences according to sex has been shown depending on the disease; however, the differences are not well-known.

This study had several limitations. First, as this was a retrospective study conducted in a single-center tertiary hospital, it may not be representative of all patients. However, as a tertiary general hospital, it is a representative institution that manages local inpatients, and it is believed that it can reflect the basic characteristics of patients. Second, our study was based on patients’ medical records, and we could not evaluate the unrecorded content. However, most of the patients’ medical records were about the basic needs of COVID-19 patients; therefore, it is unlikely that this would have a significant impact on the research. Third, it was unclear whether the patient had been vaccinated. During the first COVID-19 pandemic, vaccines were primarily administered to healthcare workers and those with risk factors. As most patients had not been vaccinated, the impact is likely to be low. In this research, 47.8% of the hospitalized COVID-19 patients were males. Males were younger, had lower CFS, and had a higher BMI than females. However, the severity of disease, invasive treatment, length of hospital stay, and in-hospital mortality did not differ between males and females. Additionally, the male sex alone was not associated with in-hospital mortality.

## 5. Conclusions

In conclusion, males were younger, had lower CFS, and had higher BMI; however, there was no significant difference in severity and in-hospital mortality. Although the severity and mortality in males are known to be higher than those in females, in this study, sex differences between males and females in COVID-19 patients were not associated with their prognosis. More studies are needed to evaluate the sex differences in the characteristics of COVID-19 and their mechanism.

## Figures and Tables

**Figure 1 jcm-12-01118-f001:**
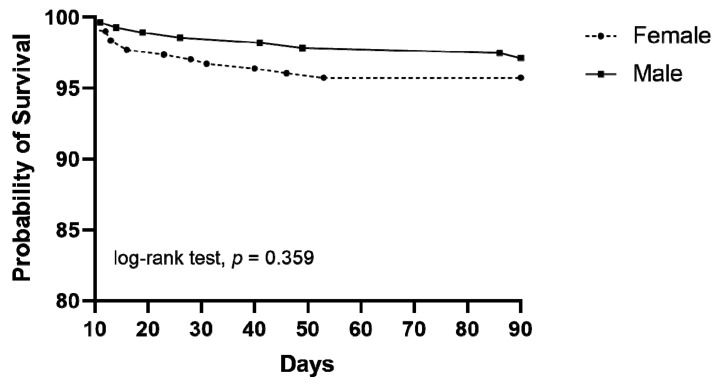
Kaplan–Meier curves of the patients according to sex (log-rank test, *p* = 0.359).

**Table 1 jcm-12-01118-t001:** Baseline characteristics and clinical features of the study population.

Characteristic	Total Patients (*n* = 584)	Female (*n* = 305)	Male (*n* = 279)	*p*-Value
Age, years	57.0 (45.0–66.0)	58.0 (48.0–68.0)	55.0 (39.0–64.0)	<0.001
Clinical frailty scale	2.0 ± 1.1	2.2 ± 1.2	1.9 ± 1.0	0.003
SOFA score	0.0 (0.0–0.0)	0.0 (0.0–0.0)	0.0 (0.0–1.0)	0.736
BMI, kg/m^2^	24.5 (22.4–26.8)	23.9 (21.7–26.2)	24.8 (22.9–27.3)	<0.001
Symptom before hospitalization				
No symptom	119 (20.4)	61 (20.0)	58 (20.8)	0.813
Fever	233 (39.9)	109 (35.7)	124 (44.4)	0.032
Cough	168 (28.8)	81 (26.6)	87 (31.2)	0.217
Myalgia	171 (29.3)	93 (30.5)	78 (28.0)	0.501
Sore throat	120 (20.5)	72 (23.6)	48 (17.2)	0.056
Dyspnea	16 (2.7)	8 (2.6)	8 (2.9)	0.857
Headache	73 (12.5)	33 (10.8)	40 (14.3)	0.199
Diarrhea	12 (2.1)	7 (2.3)	5 (1.8)	0.669
Duration of symptom before admission	4.0 (2.0–7.0)	4.0 (3.0–7.0)	4.0 (2.0–7.0)	0.320

Data are presented as median (interquartile range) or number (%), unless otherwise indicated. BMI, body mass index; SOFA, Sequential Organ Failure Assessment.

**Table 2 jcm-12-01118-t002:** Baseline comorbidities and laboratory findings of the study population.

Characteristic	Total Patients (*n* = 584)	Female (*n* = 305)	Male (*n* = 279)	*p*-Value
Underlying comorbidities				
Hypertension	181 (31.0)	93 (30.5)	88 (31.5)	0.784
COPD	15 (2.6)	4 (1.3)	11 (3.9)	0.045
Diabetes	103 (17.6)	46 (15.1)	57 (20.4)	0.090
Solid cancer	32 (5.5)	19 (6.2)	13 (4.7)	0.405
Hematologic malignancy	1 (0.2)	0 (0)	1 (0.4)	0.295
Heart failure	23 (3.9)	11 (3.6)	12 (4.3)	0.666
CKD	14 (2.4)	7 (2.3)	7 (2.5)	0.866
CVA	17 (2.9)	8 (2.6)	9 (3.2)	0.665
Liver cirrhosis	5 (0.9)	3 (1.0)	2 (0.7)	0.727
Laboratory findings				
WBC, ×10^3^/uL	4.70 (3.70–6.02)	4.34 (3.44–5.86)	5.12 (3.90–6.20)	0.003
NLR	2.41 (1.58–3.74)	2.33 (1.51–3.53)	2.49 (1.65–4.09)	0.257
Platelet, ×10^3^/uL	199 (164–243)	202 (166–245)	194 (161–241)	0.226
Total bilirubin, mg/dL	0.50 (0.35–0.67)	0.40 (0.30–0.59)	0.55 (0.41–0.73)	0.001
Albumin, g/dL	4.1 (3.7–4.3)	4.0 (3.7–4.3)	4.1 (3.7–4.4)	0.169
AST, U/L	23 (17–35)	22 (17–34)	24 (18–37)	0.129
ALT, U/L	23 (16–35)	19 (14–29)	26 (19–40)	<0.001
Creatinine, mg/dL	0.68 (0.54–0.85)	0.56 (0.46–0.67)	0.82 (0.70–0.93)	<0.001
Troponin-I. pg/mL	3.8 (2.3–6.4)	3.3 (2.3–6.1)	3.9 (2.6–6.8)	0.381
NT-proBNP, pg/mL	47.5 (21.1–116.0)	57.9 (31.3–129.6)	33.7 (13.6–92.1)	0.286
D-dimer, ng/mL	135 (78–234)	146 (93–256)	123 (59–213)	0.024
CRP, mg/dL	0.6 (0.3–2.3)	0.6 (0.3–1.5)	0.8 (0.3–3.0)	0.026
Procalcitonin, ng/mL	0.05 (0.05–0.05)	0.05 (0.05–0.05)	0.05 (0.05–0.05)	0.177
Interleukin-6, pg/mL	6.6 (3.0–20.2)	6.1 (3.1–17.9)	7.8 (2.9–26.1)	0.356
Lactic acid, mmol/L	2.2 (1.7–2.6)	2.1 (1.7–2.5)	2.2 (1.8–2.7)	0.079

Data are presented as median (interquartile range) or number (%), unless otherwise indicated. ALT, alanine aminotransferase; AST, aspartate aminotransferase; CKD, chronic kidney disease; COPD, chronic obstructive lung disease; CRP, C-reactive protein; CVA, cerebrovascular accident; NLR, neutrophil lymphocyte ratio; NLR, neutrophil lymphocyte ratio; NT-proBNP, N-terminal probrain natriuretic peptide; WBC, white blood cell.

**Table 3 jcm-12-01118-t003:** Treatment and prognosis of the study group.

	Total Patients (*n* = 584)	Female (*n* = 305)	Male *(n* = 279)	*p*-Value
Apply of vasopressors				
Vasopressors	17 (2.9)	11 (3.6)	6 (2.2)	0.296
Norepinephrine	16 (2.7)	10 (3.3)	6 (2.2)	0.404
Vasopressin	5 (0.9)	4 (1.3)	1 (0.4)	0.212
Dobutamine	4 (0.7)	3 (1.0)	1 (0.4)	0.360
O_2_ supply				
Nasal prong	101 (17.3)	54 (17.7)	47 (16.8)	0.784
HFNC	40 (6.8)	19 (6.2)	21 (7.5)	0.535
Ventilator	41 (7.0)	21 (6.9)	20 (7.2)	0.894
ECMO	12 (2.1)	7 (2.3)	5 (1.8)	0.669
Medical treatment				
Steroid	100 (17.1)	50 (16.4)	50 (17.9)	0.624
Antibiotics	115 (19.7)	60 (19.7)	55 (19.7)	0.990
Remdesivir	42 (7.2)	23 (7.5)	19 (6.8)	0.733
Severity of COVID-19				
Severe infection ^†^	48 (8.2)	29 (9.5)	19 (6.8)	0.236
Critical infection ^‡^	53 (9.1)	25 (8.2)	28 (10.0)	0.440
Arterial line	45 (7.7)	23 (7.5)	22 (7.9)	0.876
Central line	42 (7.2)	21 (6.9)	21 (7.5)	0.764
Tracheostomy	12 (2.1)	6 (2.0)	6 (2.2)	0.876
CRRT	6 (1.0)	4 (1.3)	2 (0.7)	0.477
Prognosis of patients				
In-hospital mortality	21 (3.6)	13 (4.3)	8 (2.9)	0.366
Hospital stay, days	12 (10–15)	12 (10–16)	12 (10–15)	0.537
DNR	20 (3.4)	12 (3.9)	8 (2.9)	0.479

Data are presented as median (interquartile range) or number (%), unless otherwise indicated. † Severe infection: An oxygen saturation level of 94% or less while the patient was breathing ambient air or a need for oxygen support. ‡ Critical infection: Patients showed respiratory failure, multiple organ dysfunction, or septic shock. No patients were treated with tocilizumab during the study period. CRRT, continuous renal replacement therapy; DNR, do not resuscitate; ECMO, extracorporeal membrane oxygenation; HFNC, high flow nasal cannula.

**Table 4 jcm-12-01118-t004:** Univariate and multivariate logistic regression analysis addressing the risk factors for in-hospital mortality.

	Univariate Analysis			Multivariate Analysis		
	OR	95% CI	*p*-Value	OR	95% CI	*p*-Value
Age	1.128	1.082–1.177	<0.001	1.151	1.089–1.217	<0.001
Male	0.404	0.154–1.060	0.066	1.160	0.348–3.872	0.809
BMI	1.051	0.937–1.179	0.393			
Frailty scale	1.809	1.368–2.391	<0.001	0.989	0.556–1.758	0.970
SOFA score	1.006	0.856–1.182	0.943			
Underlying disease						
COPD	3.699	0.831–16.463	0.086	3.452	0.499–23.893	0.209
Diabetes	1.759	0.677–4.574	0.247			
Initial lab						
White blood cell, ×10^3^/uL	1.379	1.195–1.593	<0.001	1.489	1.185–1.872	0.001
Creatinine, mg/dL	0.808	0.465–1.404	0.450			
CRP, mg/dL	1.132	1.069–1.199	<0.001	1.137	1.054–1.226	0.001
Lactic acid, mmol/L	1.308	0.810–2.111	0.272			

BMI, body mass index; COPD, chronic obstructive lung disease; CI, confidence interval; CRP, C-reactive protein; OR, odds ratio; SOFA, Sequential Organ Failure Assessment.

## Data Availability

The datasets used and analyzed during the current study are available from the corresponding author upon reasonable request.

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
