# Peer review of "The Influence of Sex on Characteristics and Outcomes of Coronavirus-19 Patients: A Retrospective Cohort Study"

_jcm, 2023, doi:10.3390/jcm12031118_

Round 1

Reviewer 1 Report

Despite the numerous articles published on Coronavirus-19, as it is a new disease, it is important to publish studies with epidemiological data that help in the prognosis and treatment of the viral infection.

The article, in general, is very well written, with good and up-to-date references, however, I have suggestions.

The introduction presents a literature review consistent with the proposed objectives, as well as a description of the methodology. The results are presented in the form of tables and commented appropriately to answer the objective, however, the organization of the discussion needs some improvements. Some paragraphs are repeated (example: line 157) and can be combined.

I suggest starting the paragraphs with the results obtained in the present study and then discussing them based on the current literature.

Also highlight in the conclusions which results could not be achieved. This is very important as it will help future researchers interested in the subject to identify the gaps left by the studies to propose something innovative.

Author Response

Thank you for your comments. We did our best to answer all the questions and comments raised by reviewers.

Reviewer 2 Report

1-      In paragraph 2 of the introduction, it is mentioned that most studies have shown that men are at greater risk than women. But the abstract mentions that this difference is not clear

2-      In the statistical analysis section, how to select the variables to enter the multivariate model should be mentioned

3-      In Table 3 of the results section, one p-value should be reported for the severity variable using chi-square test. The error is increased by pairwise comparisons.

4-      In Table 3, adjustments have been made for the age. In Table 4, age is included in the model along with other confounders. It is better to report the P value of age without adjustment for other variables in Table 3.

5-      In the multivariable model, there are a number of non-significant variables, it is better to remove the variables with a higher p-value from the model in order to achieve a model where the variables are significant or have a low p-value. It is important to note that for every 10 outcomes, death, you can enter one variable into the multivariate model. So, to solve this problem, it is better to remove non-significant variables with high p-value from the multivariable model in order to reach the final model.

6-      One of the main goals of your study is to compare the outcome of death in women and men. It is better to enter the sex variable in the form of interaction with influencing variables in the regression model. If the interaction term is meaningful, keep it in the model and it will greatly increase the quality of your article.

Author Response

(The authors gave the same response as above.)
